# Is There an Association between Epicardial Adipose Tissue and Outcomes after Paroxysmal Atrial Fibrillation Catheter Ablation?

**DOI:** 10.3390/jcm10143037

**Published:** 2021-07-08

**Authors:** Néfissa Hammache, Hugo Pegorer-Sfes, Karim Benali, Isabelle Magnin Poull, Arnaud Olivier, Mathieu Echivard, Nathalie Pace, Damien Minois, Nicolas Sadoul, Damien Mandry, Jean Marc Sellal, Christian de Chillou

**Affiliations:** 1Département de Cardiologie, CHRU de Nancy, F-54500 Vandœuvre-lès-Nancy, France; h.pegorersfes@chru-nancy.fr (H.P.-S.); k.benali@chru-nancy.fr (K.B.); i.magnin-poull@chru-nancy.fr (I.M.P.); a.olivier@chru-nancy.fr (A.O.); m.echivard@chru-nancy.fr (M.E.); n.pace@chru-nancy.fr (N.P.); damien.minois@gmail.com (D.M.); n.sadoul@chru-nancy.fr (N.S.); jm.sellal@chru-nancy.fr (J.M.S.); c.dechillou@chru-nancy.fr (C.d.C.); 2INSERM-IADI, U1254, F-54500 Vandœuvre-lès-Nancy, France; d.mandry@chru-nancy.fr; 3Département de Cardiologie, CHU de Saint-Etienne, 42270 Saint-Priest-en-Jarez, France; 4Département de Cardiologie, CHU de Nantes, 44000 Nantes, France; 5Département de Radiologie, CHRU de Nancy, F-54500 Vandœuvre-lès-Nancy, France

**Keywords:** paroxysmal atrial fibrillation, catheter ablation, epicardial adipose tissue

## Abstract

Background: In patients undergoing paroxysmal atrial fibrillation (PAF) ablation, pulmonary vein isolation (PVI) alone fails in maintaining sinus rhythm in up to one third of patients after a first catheter ablation. Epicardial adipose tissue (EAT), as an endocrine-active organ, could play a role in the recurrence of AF after catheter ablation. Objective: To evaluate the predictive value of clinical, echocardiographic, biological parameters and epicardial fat density measured by computed tomography scan (CT-scan) on AF recurrence in PAF patients who underwent a first pulmonary vein isolation procedure using radiofrequency (RF). Methods: This monocentric retrospective study included all patients undergoing first-time RF PAF ablation at the Nancy University Hospital between March 2015 and December 2018 with one-year follow-up. Results: 389 patients were included, of whom 128 (32.9%) had AF recurrence at one-year follow-up. Neither total-EAT volume (88.6 ± 37.2 cm^3^ vs. 91.4 ± 40.5 cm^3^, *p* = 0.519), nor total-EAT radiodensity (−98.8 ± 4.1 HU vs. −98.8 ± 3.8 HU, *p* = 0.892) and left atrium-EAT radiodensity (−93.7 ± 4.3 HU vs. −93.4 ± 6.0 HU, *p* = 0.556) were significantly associated with AF recurrence after PAF ablation. In multivariate analysis, previous cavo-tricuspid isthmus (CTI) ablation, ablation procedure duration, BNP and triglyceride levels remained independently associated with AF recurrence after catheter ablation at 12-months follow-up. Conclusion: Contrary to persistent AF, EAT parameters are not associated with AF recurrence after paroxysmal AF ablation. Thus, the role of the metabolic atrial substrate in PAF pathophysiology appears less obvious than in persistent AF.

## 1. Introduction

Atrial fibrillation (AF) is the most common cardiac arrhythmia affecting approximatively 33.5 million persons worldwide [1]. Current guidelines recommend catheter ablation (CA) in order to maintain sinus rhythm and improve quality of life in symptomatic patients in whom drugs have already failed [2]. Currently, pulmonary vein isolation (PVI) is the cornerstone of the ablation strategy in patients with paroxysmal atrial fibrillation (PAF). This is a well-established treatment for the prevention of PAF recurrence. The 12-month success rate is approximately 65% after a first procedure and 80% after multiple procedures [3,4,5,6]. Recently, it has been shown that first-line PAF ablation is superior to antiarrhythmic therapy in terms of recurrence, but also in terms of symptom improvement, physical capacity, and quality of life, reinforcing the place of catheter ablation [7,8,9]. Thus, determining the factors contributing to recurrence after a catheter ablation becomes essential. Prognostic models, which combine several predictors (such as left atrium volume, sex, age, coronary artery disease) to generate an individualized risk estimate, have been developed for prediction of AF recurrence after catheter ablation [10]. None of them has proven to be effective, especially in PAF, where only PV reconnections have proven to be a significant predictive factor for AF recurrences.

Epicardial adipose tissue (EAT) serves as a biologically active organ with important endocrine and inflammatory function [11,12]. An accumulating body of evidence suggests that EAT is associated with the initiation, perpetuation, and recurrence of AF, especially in case of persistent AF (PersAF) [13,14,15]. Quantitative and qualitative evaluation of EAT, aided by the development of imaging techniques, is of growing interest [16]. The role of EAT in recurrence after PAF catheter ablation has not been clearly elucidated.

The aim of this study was to evaluate the predictive value of clinical, echocardiographic, biological parameters and EAT characteristics measured by computed tomography scan (CT-scan) on AF recurrence in PAF patients after a first radiofrequency (RF) PVI.

## 2. Materials and Methods

### 2.1. Study Population

Consecutive adult patients with symptomatic drug-refractory PAF referred for a first RF catheter ablation between April 2015 and December 2018 in Nancy University Hospital were included. Preprocedural CT-scan was routinely performed before AF catheter ablation. AF was considered to be paroxysmal if it terminated spontaneously or with intervention within 7 days of onset.

Inclusion criteria were: first procedure of catheter ablation for PAF with CT scan, transthoracic echocardiography and transesophageal echocardiography before the procedure and >12-month follow-up. Exclusion criteria were: prior AF catheter ablation and cryo-balloon ablation, LA linear lesions, LA defragmentation and intervention aborted due to a procedural complication. All patients were adults and provided written informed consent for the CA and all procedures were in line with current guidelines (see Figure 1).

### 2.2. Echocardiography

Transthoracic and transesophageal echocardiography were performed within 24 h before intervention using a Vivid S6 cardiovascular ultrasound system (General Electric, Horten, Norway). The following data were collected: left ventricular ejection fraction (biplane Simpson’s method), left atrial (LA) surface area (apical four-chamber view at end-systole), indexed LA volume (biplane area-length method at end-systole), LA diameter (parasternal long axis view), diastolic function according to the 2016 ASE recommendations (E/A, lateral e’ velocity, average E/e’, TR velocity), measurement of the interventricular septum at end-diastole.

### 2.3. Cardiac Computed Tomography

A 256-slice multidetector cardiac scanner (Revolution CT, General Electric) with iodinated contrast product injection was performed before the procedure to assess PVs and LA anatomy, and to check the absence of thrombus in the left atrial appendage. Cardiac CT angiography, with electrocardiographic gating, was acquired using a scanner allowing up to 16 cm of detector coverage, so that the whole heart could be captured in a single heartbeat. ECG-gated acquisitions were obtained during end-systole (approximately 40% of R–R interval). The injection protocol included an initial contrast injection of 50 mL (iodine concentration of 350 mg/mL) at the rate of 5 mL/s followed by 40 mL of saline at the rate of 4 mL/s. The acquisitions settings were 100 kV tube potential, 500 mA tube charge, 0.28 s rotation. Images were reconstructed with a slice thickness of 0.625 mm. Total EAT volume and density were assessed using a semi-automatic procedure with the CardIQ Xpress 2.0 v post-processing software on an Advantage workstation 4.7 v (General Electric) (see Figure 2).

### 2.4. Cardiac CT Image Analysis

Manual contouring of the fibrous pericardium was performed on axial planes, for every 10 mm, from the pulmonary artery bifurcation to the diaphragm (see Figure 3). EAT was detected by assigning an attenuation threshold from −50 to −250 Hounsfield units (HU) to fat [17]. After three-dimensional reconstruction, volume (in cm^3^), mean density (in HU) ± standard deviation (SD) were automatically calculated by the software. In order to assess left atrial (LA)-EAT density, three areas were identified: superior left region (SL), inferior left region (IL) and inferior right region (IR), near the pulmonary vein ostia. A circular region of interest (ROI) of 20 mm^2^, or the largest possible size < 20 mm^2^, was manually drawn and placed in each region (see Figure 4). LA density was calculated as the average of all regional densities. Data were evaluated by one operator, blinded to clinical outcomes.

Fibrous pericardium was manually traced, on axial planes, for every 10 mm, from the pulmonary artery bifurcation to the diaphragm. EAT was detected by assigning an attenuation threshold from −50 to −250 Hounsfield units (HU) to fat.

In order to assess LA-EAT density, three areas were identified: superior left region (SL), inferior right region (IL) and inferior left region (IR), near the PV ostia. A circular ROI of 20 mm^2^, failing the largest possible size, was manually drawn and placed in each region. LA density was calculated as the average of all regional densities.

### 2.5. Atrial Fibrillation Ablation

AF ablation procedures were performed under local anesthesia and conscious sedation. Two catheters were advanced from the right femoral vein to the LA through transseptal puncture, under fluoroscopic guidance, a 10-pole circular mapping catheter (Lasso, Biosense Webster, Diamond Bar, CA, USA) and a 3.5-mm externally irrigated-tip ablation catheter (3.5-mm tip, ThermoCool, Biosense Webster, Diamond Bar, CA, USA/3.5-mm tip, Flexability, St. Jude Medical, St. Paul, MN, USA). A steerable quadripolar catheter (Xtrem, SORIN Group, Clamart, France) was placed into the coronary sinus and used as an electro anatomical mapping reference.

A 3-dimensional navigation system (CARTO^®^, Biosense Webster, Inc, Irvine, CA, USA or EnSite NavX system St. Jude Medical, St Paul, MN, USA) was used to create a 3-dimensional electro-anatomical map of the LA, which was integrated with computed tomography of the LA. PVI was performed with radiofrequency energy in a point-by-point wide area circumferential ablation (two by two PVI) pattern using a Thermocool SmartTouch irrigated tip CF-sensing ablation catheter (Biosense Webster, Inc., Irvine, CA, USA) or a FlexAbility^TM^ Ablation Catheter (St. Jude Medical, St. Paul, MN, USA) introduced via a non-steerable sheath. The point-by-point circumferential lesion sets were created while navigating the catheter under the guidance of a 3-D electro-anatomical mapping system. During ablation, computerized LA reconstruction and mapping was conducted using the CARTO^®^ mapping system (Biosense Webster, Inc. Irvine, CA, USA) or the EnSite NavX system (St. Jude Medical, St Paul, MN, USA). Contiguous lesions were performed. Each lesion was respectively guided by ablation index (AI) targets using 450 at the roof and anterior walls and 350 at the posterior and inferior walls or lesion size index (LSI) targets using 5.5 at the roof and anterior walls and 4.5 at the posterior and inferior wall. RF pulses were delivered by using a 550-kHz RF Stockert-Cordis generator and the ablation catheter, in a power-controlled mode, with RF energy up to 30 Watts at the anterior part of the veins and 25 Watts at their posterior part.

### 2.6. Endpoints and Data Collection

The objective was to assess the association between clinical, biological, echocardiographic and EAT characteristics and PAF recurrence after radiofrequency catheter ablation. The medical software DxCare^®^ was used to collect all patients’ data needed for the study. We evaluated: demographic and physical features, comorbidities, treatments, AF history, laboratory analysis, electrocardiogram features at the beginning of the procedure and at hospital discharge, and ablation procedure modalities.

### 2.7. Follow-Up and AF Recurrence Assessment

AF recurrences were assessed after a 3-month blanking period, and defined as ≥1 AF episode recorded during a 12 lead ECG or ≥1 AF episode lasting ≥30 s documented by Holter monitoring. All patients underwent an electrocardiographic evaluation before discharge from hospital. Arrhythmia monitoring included clinical evaluation, 12-lead electrocardiogram in case of symptom recurrence, and systematic 24-h Holter monitor recording by the referring cardiologist at months 3, 6 and 12. Our investigative team was unaware of the follow-up assessment outcomes.

The continuation or initiation of anti-arrhythmic drug therapy post-procedure and at 3 months was left to the referring physician preference. Successful ablation was defined as the absence of a documented episode of AF with or without anti-arrhythmic at one year after the procedure.

### 2.8. Statistical Analysis

Statistical analyzes were performed using IBM SPSS^®^ version 26 software. Continuous variables were reported as mean ± standard deviation if normally distributed or median with interquartile range (IQR) if not. Categorical variables were expressed as frequencies with percentages. The Student’s *t*-test or Mann-Whitney U-test (depending on whether the values were normally distributed) allowed comparison of continuous variables while comparison of percentages was performed using the Pearson’s chi-squared test.

Simple logistic regression analysis was conducted to identify the relationship between baseline characteristics and recurrence. Variables with *p* < 0.100 were included in multivariate regression analysis. Because our study was focused on EAT, we planned to include EAT parameters in the multivariate analysis whatever the result in univariate analysis. A *p*-value < 0.05 was considered statistically significant.

## 3. Results

### 3.1. Baseline Characteristics

Between March 2015 and December 2018, 466 patients underwent their first CA for PAF. 77 patients were excluded (seven patients without CT scan; five aborted interventions due to intra-procedural tamponade [3], stroke [1], and intolerable pain [1], as well as 65 patients that underwent LA defragmentation). The population was divided into two groups, AF recurrence and successful CA, after 12-months follow-up. Patients’ characteristics are summarized in Table 1.

#### Cardiac Computed Tomography Features

Total EAT volume and density were 90.5 ± 39.4 cm^3^ and −98.9 ± 3.9 HU, respectively. LA-EAT density was −93.5 ± 5.5 HU. EAT density of the LS region, LI region and RI region were respectively −93.8 ± 12.1, −93.4 ± 7.6 and −93.3 ± 7.6 HU.

### 3.2. Ablation Results

During the 12 months’ follow-up, 128 patients (32.9%) presented recurrence of AF.

#### Independent Predictors of AF Recurrence

Among the variables, only dyslipidemia, CHADsVASc score, previous CTI ablation, AF during procedure, RF time, presence of dyspnea, triglycerides level and BNP were significantly associated with AF recurrences after univariate analysis (Table 2).

The associations with AF recurrence at 12 months after CA remained significant following multivariate analysis (Table 3) for previous CTI ablation (*p* = 0.013, OR 2.43, 95% CI 1.22–4.85), RF time per ten minutes (*p* = 0.033, OR 1.20, 95% CI 1.12–1.30), BNP per 100 pg/mL (*p* = 0.019, OR 1.35, 95% CI 1.22–1.49) and triglycerides level (*p* = 0.047, OR 1.54, 95% CI 1.02–2.26). No association was found between AF recurrence and EAT, either total volume, total density or LA density with, respectively, *p* = 0.968, OR 1.00, 95% CI 0.99–1.01, *p* = 0.432, OR 1.02 95% CI (0.96–1.1) and *p* = 0.771, OR 1.06 95% CI (0.96–1.16).

## 4. Discussion

### 4.1. Main Results

The main finding of our study is that EAT parameters are not associated with the recurrence of PAF at one-year follow-up after a first CA. On the other hand, previous CTI ablation, longer RF time, high BNP and triglyceride levels appear to be factors associated with AF recurrence after the first CA procedure in patients treated for PAF.

### 4.2. EAT and PAF Recurrence after CA

To our knowledge, this is the first study designed to examine the association between EAT, assessed by cardiac CT, solely in a PAF population and not in a mixed population of PersAF and PAF. Numerous studies have demonstrated a relationship between EAT and the risk of AF recurrence after ablation [18]. These studies included mostly PersAF. However, in subgroup analyses according to the type of AF, the results differed between paroxysmal and persistent AF [19]. In these studies, EAT appeared to have a prognostic value for AF recurrences only in the persistent AF subgroup, without significant association in the PAF subgroup. Our results are consistent with these previous findings, suggesting the existence of different pathophysiological mechanisms between paroxysmal and persistent AF regarding EAT involvement.

EAT volume has been shown to be bigger in persistent AF compared with PAF [20,21,22]. The volume and density of the EAT, reflecting its secretory activity, is a predictive factor of recurrence after ablation of PersAF. Studies on patients with AF undergoing cardiac CT and electroanatomic mapping showed that the presence of EAT was associated with lower bipolar voltage and more electrogram fractionation as electrophysiologic substrates for AF [23]. These changes underlie the pathophysiology of PersAF and are prime targets during CA. As the EAT is in contact with the atrial epicardium, the question of hybrid ablation is justified. A recent study by Vroomen et al. could not confirm that EAT-V was predictive of recurrence of atrial fibrillation in patients undergoing a hybrid AF ablation. It might be interesting to look at EAT density in order to target areas of interest during hybrid ablation.

In contrast, the majority of PAF patients can be treated by eliminating PV triggers, and PVI is enough, with only a minor role for the previously described substrate and thus of the associated EAT.

However, a recent study published by El Mahdiui et al. showed an association between the EAT density of the LA posterior wall and the risk of recurrence after ablation of any type of AF [24]. Batal et al. had previously reported that only posterior EAT thickness was associated with AF burden [21]. The importance of posterior EAT in PAF could be explained by the effect of direct local EAT secretion on the PVs. Our study focused on the fat density of the entire LA. It might be interesting to look specifically at the posterior EAT in our population.

Moreover, the recent study of Zaman et al. found that patients with PAF despite prior PVI show electrical substrates that resemble PersAF more closely than patients with PAF undergoing their first ablation [25]. In this study, the redo group had PVI only during the first CA. In the absence of LA defragmentation, a causal relationship between the first ablation and the presence of atrial substrate more similar to that of PersAF would be difficult to understand. Notably, these subgroups of PAF are indistinguishable by structural indices. It could be interesting to compare the EAT characteristics before the first and second CA procedures, looking for criteria that may indicate a heterogeneous population of PAF that may overlap with persistent AF.

### 4.3. Previous CTI Ablation and PAF Recurrence after CA

The association of AF and common-type atrial flutter has been previously described and is very frequent. In our study, we found that a previous CTI ablation is associated with a higher risk of recurrence of AF after a catheter ablation, with a two-fold relative risk of recurrence compared to patients without previous common-type atrial flutter.

Moreira et al. demonstrated that in patients with coexisting PAF/common-type atrial flutter, CTI ablation and PVI were used successfully to treat sustained common-type atrial flutter but appeared insufficient to prevent recurrences of AF. In this population, the very existence of a common-type atrial flutter can be a sign that non-PV substrates are involved [26].

### 4.4. RF Time and PAF Recurrence after CA

Increase of ten minutes of RF application was associated with a 1.2-fold increase in risk of recurrence of AF after a catheter ablation. This can suggest that PVI was not easy to achieve in the case of longer RF time. Since PVI is a cornerstone of AF management, difficulties in achieving this outcome may be linked to an increased risk of recurrence. An increased RF time can also be caused by larger pulmonary veins ostia. Unfortunately, we did not collect this data. This criterion has never been specifically studied and is not included in the risk scores for AF recurrence after ablation.

### 4.5. BNP, Lipid Profile and Recurrence after CA

Studies suggested that elevated baseline BNP level is associated with AF recurrence after CA, suggesting that BNP could be a useful biomarker for predicting AF recurrence [27,28]. Our study is consistent with this result, describing that an increase of 100 pg/mL in the BNP was associated with a 1.2-fold increase in risk of recurrence of AF after a first PAF CA procedure.

AF is characterized by a loss of atrial contraction, leading to an increase of the LA volume and atrial stretch, promoting activation of the natriuretic system and the secretion of ANP and BNP, both of which are mainly produced in the atrium [28]. However, this effect seems to be more prominent in patients with persistent AF in whom BNP levels are significantly higher.

In our PAF population, other mechanisms are probably involved. On one hand, the increase of BNP level may reflect an electrophysiological disturbance, which could trigger AF by intracellular calcium overload, reduced conduction velocity and increased dispersion of the refractory period [28]. On the other hand, BNP may be arrhythmogenic by inhibiting sympathetic activity and potentiating vagal activity through the cGMP pathway [29].

Augmentation of 1 g/L of triglyceride level was associated with a 1.74-fold increase in risk of recurrence of AF after a catheter ablation. Metabolic syndrome, which includes high triglyceride levels, is associated with higher recurrence rates after CA, especially in case of PersAF. Metabolic syndrome is associated with PAF, but its role in recurrence is less clear [30,31]. Why metabolic syndrome markers failed to predict outcomes in patients with PAF is not known. We can speculate that their lower frequency in the PAF population may lead to a failure to show the association between recurrence after PAF catheter ablation and metabolic syndrome. Indeed, in our population the mean TG level was 1.11 g/L, which is well below the threshold value for metabolic syndrome. Under these conditions, it might be interesting to compare the characteristics of the EAT in our population according to the presence or absence of a metabolic syndrome.

### 4.6. Study Limitations

The present study has several limitations that must be considered. First, it is a retrospective monocentric study, therefore there is a non-negligible risk of selection bias. In order to assess left atrial (LA)-EAT density, three areas were identified: superior left region (SL), inferior left region (IL) and inferior right region (IR), near the pulmonary vein ostia. Other authors have decided to focus on the posterior wall of the LA. We decided to make another choice because we were interested only in the PAF where the pulmonary veins are at the center of the physiopathology. Finally, the current study evaluated the outcomes 12 months after ablation. Further studies will be necessary to evaluate the impact of EAT on late recurrence beyond 12 months. 

## 5. Conclusions

Contrary to persistent AF, EAT parameters are not associated with AF recurrence after paroxysmal AF ablation. Thus, the role of the metabolic atrial substrate in PAF pathophysiology appears less obvious than in persistent AF.

## Figures and Tables

**Figure 1 jcm-10-03037-f001:**
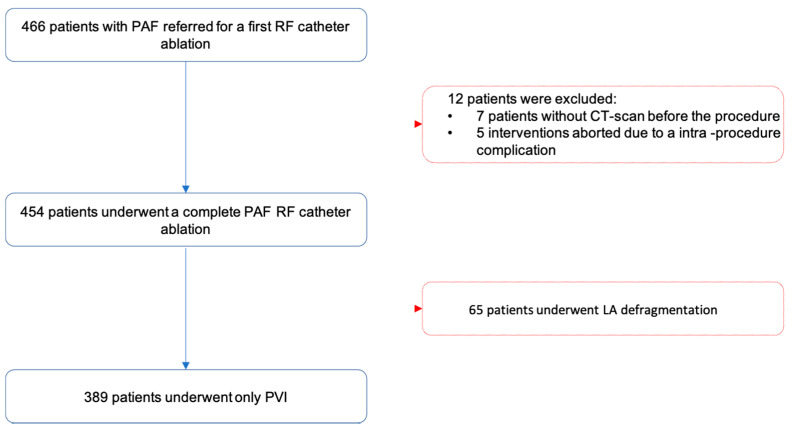
Flowchart for inclusion of patients in the study.

**Figure 2 jcm-10-03037-f002:**
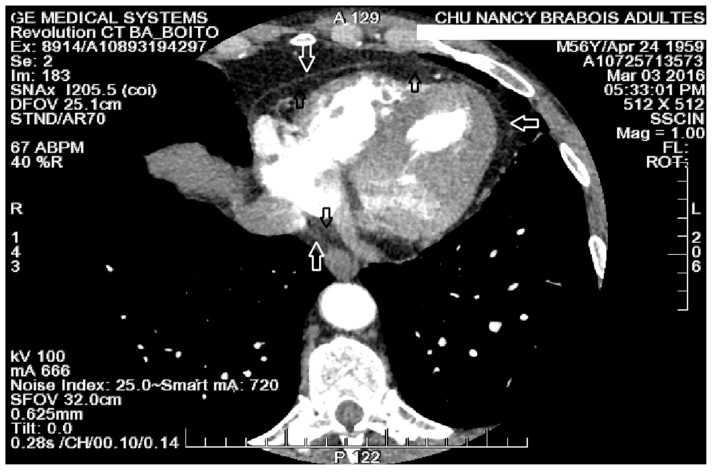
EAT measurement on a multiplanar reformatted CT image of the heart. Fibrous pericardium (white arrows) and epicardial fat (black arrows).

**Figure 3 jcm-10-03037-f003:**
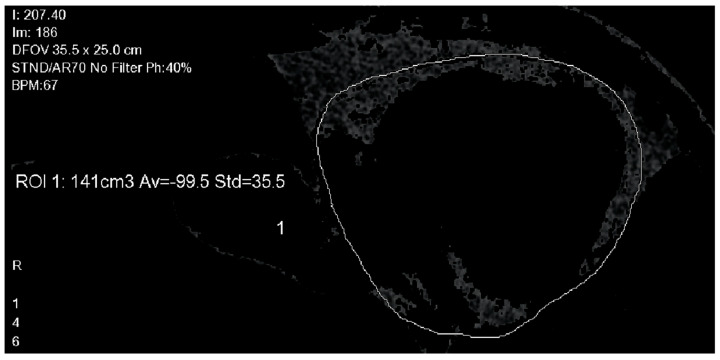
EAT measurement: contouring of the fibrous pericardium.

**Figure 4 jcm-10-03037-f004:**
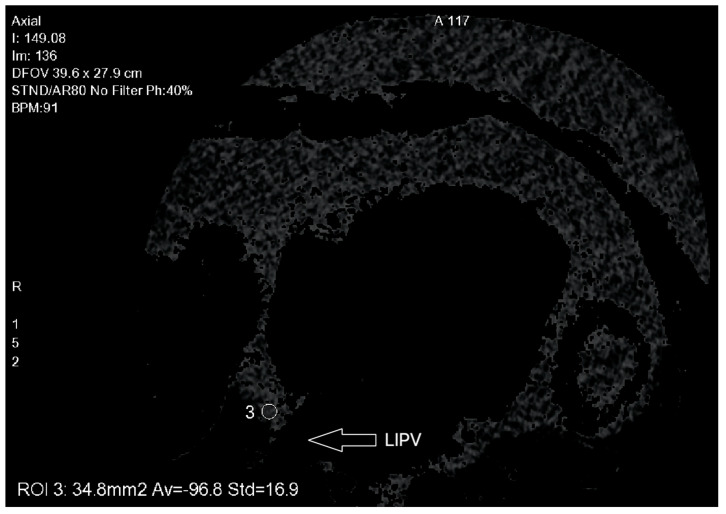
EAT measurement: Inferior left region (IL) near the LIPV (left inferior pulmonary vein) ostium.

**Table 1 jcm-10-03037-t001:** Baseline data of the overall study cohort and 1-year-follow-up (divided into patients with and without AF recurrence during 1-year-follow-up).

Variable	Total Population (*n* = 389)	AF Recurrence (*n* = 128)	Successful Procedure (*n* = 261)	*p* Value
Demographics				
Age—years	58.1 ± 11.1	57.1 ± 12	58.6 ± 10.7	0.343
Male sex—no. (%)	256 (65.8)	83 (64.8)	173 (67.6)	0.778
Co-morbidities				
HFrEF—no. (%)	25 (6.5)	5 (3.9)	20(7.7)	0.125
Hypertension—no. (%)	156 (40.1)	47 (36.7)	109 (41.8)	0.340
Diabetes—no. (%)	28 (7.2)	8 (6.3)	20 (7.7)	0.612
Dyslipidemia—no (%)	104 (26.4)	27 (21.1)	77 (29.5)	0.078
CHA2DS2-VASc Score	1.3 ± 1.3	1.2 ± 1.2	1.4 ± 1.3	0.274
Active or previous smoking—no. (%)	85 (21.9)	29 (22.7)	56 (21.5)	0.788
Obstructive sleep apnea—no. (%)	38 (9.8)	9 (7.0)	29 (11.1)	0.403
COBP—no (%)	8 (2.1)	5 (3.9)	3 (1.1)	0.072
Obesity—no (%)	82 (22.1)	30 (23.4)	52 (19.9)	0.436
Stroke —no (%)	20 (5.1)	5 (3.9)	15 (5.7)	0.476
Previous CTI ablation—no (%)	53 (13.6)	25 (19.5)	28 (10.7)	0.017
AF history and management				
Time between diagnosis and ablation—months	30.1 ± 24.2	27.6 ± 23.0	31.1 ± 24.6	0.263
Number of AF episode ≥ 1/24 h—(%)	93 (23.9)	30 (23.5)	63 (23.9)	0.308
AF at the beginning of procedure—no (%)	22 (5.7)	8 (6,5)	14 (5.6)	0.722
AF during procedure—no (%)	149 (36.1)	54 (42.2)	86 (32.9)	0.079
AF at the end of procedure no (%)	21 (5.4)	9 (7.0)	12 (4.6)	0.318
RF time (minutes)	34.4 ± 15.6	36.8 ± 15.9	33.2 ± 15.1	0.033
AF recurrence (first year)	153 (33.7)			
Medication				
B-blocker—no (%)	211 (54.2)	69 (53.9)	142 (54.4)	0.926
ACEI—no (%)	57 (14.7)	15 (11.7)	42 (16.1)	0.252
ARB—no (%)	71 (18.3)	23 (18.0)	48 (18.4)	0.919
MRA—no (%)	17 (4.4)	3 (2.3)	14 (5.4)	0.171
Statines—no (%)	78 (23.6)	20 (15.6)	58 (22.2)	0.088
Physical features				
Body Mass Index—kg/m^2^	27.1 ± 4.7	27.7 ± 5.2	26.9 ± 4.8	0.255
NYHA functional class—no. (%)				
II/III/IV	89 (22.8)	30 (23.4)	59 (22.6)	0.089
Laboratory analysis				
Total cholesterol level—g/L	2.36 ± 0.78	1.89 ± 0.39	2.59 ± 0.39	0.509
Triglycerides level—g/L	1.10 ± 0.73	1.17± 0.92	1.10 ± 0.62	0.168
LDL level—g/dl	1.15 ± 0.32	1.16 ± 0.33	1.14 ± 0.32	0.693
HDL level—g/dl	0.50 ± 0.13	0.51 ± 0.13	0.50 ± 0.13	0.738
Creatinine—umol/L	84.7 ± 38.2	82.7 ± 22.8	85.7 ± 34.1	0.632
CRP—mg/L	3.8 ± 8.8	3.8 ± 7.0	3.8 ± 9.7	0.962
Fibrinogen—g/L	3.2 ± 0.8	3.2 ± 0.8	3.1 ± 0.8	0.372
BNP—pg/mL	83.9 ± 107.3	99.9± 133	77.7± 88.8	0.028
Echocardiography features				
LVEF—%	59.7 ± 7.4	59.4 ± 7.2	59.9 ± 7.6	0.524
Left atrial enlargement—no (%)	168 (46.2)	58 (45.3)	110 (42.1)	0.638
Left atrial surface area—cm^2^	20.2 ± 5.1	20.5 ± 5.3	20.0 ± 5.0	0.516
Left atrial volume—ml/m^2^	34.2 ± 11.2	34.7 ± 11.6	34.0 ± 10.9	0.527
Left ventricular hypertrophy—no (%)	34 (9)	13 (10.2)	21 (8)	0.480
Left ventricular diastolic dysfunction—no (%)	66 (17.0)	23 (17.6)	43 (16.4)	0.830
CT features				
Total EAT Volume—cm^3^	90.5 ± 39.4	88.6 ± 37.2	91.4 ± 40.5	0.519
Total-EAT Density—HU	−98.9 ± 3.9	−98.8 ± 4.1	−98.8 ± 3.8	0.892
LA-EAT Density—HU	−93.5 ± 5.5	−93.7 ± 4.3	−93.4 ± 6.0	0.556
LA EAT (LS GP) Density—HU	−93.8 ± 12.1	−94.1 ± 6.9	−93.7± 14.0	0.784
LA EAT (RI GP) Density—HU	−93.4 ± 7.6	−92.7 ± 7.4	−93.7 ± 7.7	0.198
LA EAT (LI GP) Density—HU	−93.3 ± 7.6	94.4 ± 7.3	−92.8 ± 7.7	0.175

COBP: chronic obstructive broncho-pneumopathy; MRA: mineralocorticoid receptor antagonist.

**Table 2 jcm-10-03037-t002:** Univariate analysis of AF recurrence after CA after 12-months Follow-Up.

Variables	Odds Ratio (95% CI)	*p* Value
Demographics		
Age-years	0.99 (0.97~1.01)	0.232
Male sex	0.93 (0.60~1.46)	0.779
Co-morbidities		
HFrEF	0.49 (0.2~1.24)	0.133
Hypertension	0.81 (0.52~1.25)	0.341
Diabete	0.80 (0.34~1.88)	0.613
Dyslipidemia	0.64 (0.39~1.05)	0.080
CHA2DS2-VASc Score	0.85 (0.72~1.0)	0.056
Active or previous smoking	1.07 (0.65~1.78)	0.788
Obstructive sleep apnea	0.71 (0.32~1.58)	0.405
Stroke	0.52 (0.18~1.54)	0.24
Previous CTI ablation	2.02 (1.19~3.68)	0.019
AF history and management		
Time between diagnosis and ablation—months	0.99 (0.98~1.00)	0.223
Number of AF episodes ≥ 1/24 h	1.03 (0.78 ~1.36)	0.847
AF at the beginning of procedure	1.2 (0.48~2.88)	0.722
AF during procedure	1.48 (1.0~2.3)	0.08
AF at hospital discharge	1.57 (0.64~3.83)	0.322
RF time—minutes	1.01 (1.00~1.02)	0.035
Medication		
Beta-blocker	0.98 (0.64~1.5)	0.926
ACEI	0.69 (0.37~1.3)	0.254
ARB	0.97 (0.56~1.68)	0.919
MRA	0.42 (0.12~1.5)	0.183
Statine	0.99 (0.69~1.42)	0.957
Physical features		
Body Mass Index—kg/m^2^	1.03 (0.98~1.03)	0.256
NYHA functional class		
NYHA > I	1.8 (1.01~3.0)	0.085
Laboratory analysis		
Total cholesterol level—g/L	0.99 (0.92~1.05)	0.983
Triglycerides level—g/L	1.32 (1.02~1.71)	0.036
LDL level—g/L	1.12 (0.57~2.18)	0.741
HDL level—g/dl	1.33 (0.25~6.9)	0.737
CRP—mg/L	1.00 (0.98~1.02)	0.941
Creatinine—umol/L	1.00 (0.99~1.01)	0.64
Fibrinogen—g/L	1.15 (0.89~1.49)	0.292
BNP—pg/mL	1.00 (1.00~1.00)	0.017
Echocardiography features		
Left ventricular ejection fraction	1.00 (0.97~1.02)	0.523
Left atrial surface area—cm^2^	1.02 (0.98~1.07)	0.321
Left atrial volume—mL/m^2^	1.01 (0.99~1.03)	0.526
Left atrial dilatation	1.19 (0.80~1.78)	0.408
Left ventricular diastolic dysfunction	1.04 (0.96~1.12)	0.392
Left ventricular hypertrophy	1.07 (0.55~2.06)	0.912
CT features		
Total EAT Volume—cm^3^	1.00 (0.99~1.00)	0.518
Total EAT Density—HU	1.00 (0.95~1.06)	0.892
Total LA-EAT Density—HU	0.99 (0.94~1.03)	0.556
LA EAT (LS GP) Density—HU	1.01 (0.99~1.02)	0.784
LA EAT (RI GP) Density—HU	1.02 (0.99~1.05)	0.198
LA EAT (LI GP) Density—HU	0.97 (0.95~1.00)	0.108

**Table 3 jcm-10-03037-t003:** Multivariate analysis of AF recurrence after CA after 12-months Follow-Up.

Variables	Odds Ratio (95% CI)	*p* Value
NYHA > I	1.77 (1.04~3)	0.058
Dyslipidemia	0.58 (0.33~1.11)	0.106
CHADs VASc	0.5 (0.14~1.71)	0.268
Previous CTI ablation	2.43 (1.22~4.85)	0.013
AF during procedure	1.35 (0.8~2.29)	0.264
RF time—10 min	1.2 (1.11~1.3)	0.033
Triglycerides level—g/L	1.54 (1.02~2.26)	0.047
BNP—100 pg/mL	1.35 (1.22~1.49)	0.019
Total EAT Volume—cm^3^	1.0 (0.99~1.01)	0.968
Total EAT Density—HU	1.02 (0.96~1.10)	0.432
Total LA-EAT Density—HU	1.06 (0.96~1.16)	0.771
LA EAT (LS GP) Density—HU	0.98 (0.94~1.02)	0.249
LA EAT (RI GP) Density—HU	1.02 (0.99~1.04)	0.780
LA EAT (LI GP) Density—HU	0.97 (0.94~1.00)	0.067

## Data Availability

The data presented in this study are available on request from the corresponding author.

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
