# Peer review of "Is There an Association between Epicardial Adipose Tissue and Outcomes after Paroxysmal Atrial Fibrillation Catheter Ablation?"

_jcm, 2021, doi:10.3390/jcm10143037_

Round 1
Reviewer 1 Report
Please see attached file.

Author Response
To begin with, I would like to thank you for taking the time to carefully review our work. Your help has been invaluable.
The minor and major changes have all been made. We have added a comment about the potential effect of a first ablation on the substrate:
On lines 272-276: Moreover, a recent study of Junaid A.B. Zaman et al. found that patients with PAF despite prior PVI show electrical substrates that resemble PersAF more closely than patients with PAF undergoing their first ablation25. In this study, the redo group had PVI only during the first CA. In the absence of LA defragmentation, a causal relationship between the first ablation and the presence of atrial substrate more similar to that of PersAF would be difficult to understand.
We hope this is appropriate.
Yours faithfully

Reviewer 2 Report
Thank you for your paper,
in my opinion the conclusion linking EAT and persist Afib is very interesting and need an improvement of the paper:
- is there a relation to coronary disease? And Obesity? (well known risk factor for AF also in Guidelines)
- Is there a measurement of Homocystein? (risk factor for recurrence)
- improve the discussion with a suggestion for hybrid procedures in case of persist AFib (epicardial + endocardial). EAT - Epicardial ablation
Author Response
To begin with, I would like to thank you for your review.
Concerning obesity: we had studied this criterion in univariate analysis and there was no association with recurrence. Since this criterion was redundant with BMI we chose to keep only this second criterion since the proportion of obese was minimal in our PAF population (less than 25%). Do you think this is relevant? Or would it be better to put forward the criterion "obesity" rather than "BMI"?
Regarding homocysteine and CAD, being a retrospective study, these data were not available. We also regret this.
Finally, we thank you for your suggestion on hybrid ablation and we have been able to add to the discussion on this subject
Lines 252-261: The volume and density of the EAT, reflecting its secretory activity, is a predictive factor of recurrence after ablation of PersAF. Studies on patients with AF undergoing cardiac CT and electroanatomic mapping showed that the presence of EAT was associated with lower bipolar voltage and more electrogram fractionation as electrophysiologic substrates for AF23. These changes underlie the pathophysiology of PersAF and are prime targets during CA. As the EAT is in contact with the atrial epicardium, the question of hybrid ablation is justified. A recent study by Mindy Vroomen et al. could not confirm that EAT-V was predictive of recurrence of atrial fibrillation in patients undergoing a hybrid AF ablation. It might be interesting to look at EAT density in order to target areas of interest during hybrid ablation.
Yours faithfully
